# The Role of IPC in Setting Standards for Experimental Evaluation in Planning Research (Discussion Topic)

**Michael Katz**
IBM Research
Yorktown Heights, NY, USA
michael.katz1@ibm.om

**Silvan Sievers**
University of Basel
Basel, Switzerland
silvan.sievers@unibas.ch

Designing an experiment, specifically choosing a baseline for comparison as well as the benchmark set, that adequately evaluates the performance of a new suggested technique is a challenging task. Arguably, planner performance in a competition may affect the choice of a baseline. Further, the benchmark set choice may be biased towards the more recent domains, or, sometimes, the other way around, the most recent domains might be ignored.

We observed that the collection of domains introduced in a competition often aims at fostering research into a specific issue (e.g., general costs, conditional effects, large number of parameters) and thus the best performers in a competition, while excelling on these domains, might exhibit worse performance on other domains. This can be problematic and a misleading conclusion, given that researchers often seem to conclude from the latest IPC results what is the state of the art in planning.

We suggest to extend the experimental evaluation performed at future IPCs to include as many available domains as possible and computationally feasible (e.g., possibly a diverse subset of domains from previous IPCs). This extension could be an *additional* evaluation to keep the current setting that evaluates planners exclusively on new, unseen domains, which we still deem important to avoid domain-dependent overfitting of planners.

Furthermore, we suggest to create various suites according to task features and present additional performance information for existing relevant planners on each such suite. That might require to maintain such features, as well as label planning tasks according to these features, preferably in an automated manner. Such information can help, among other, in identifying the current state-of-the-art for specific fragments of planning, as well as the relevant benchmark set, making designing an experiment easier, mitigating possible biases.