# OpenReview forum: "The Role of IPC in Setting Standards for Experimental Evaluation in Planning Research"
_icaps-conference.org/ICAPS/2019/Workshop/WIPC_

### Official Review · AnonReviewer2 · 2019-04-25
**The Role of IPC in Setting Standards for Experimental Evaluation in Planning Research**

**Rating:** 7
**Confidence:** 4

**Review:**

This abstract discusses the role of the IPC in driving the direction of planning research.  Specifically the trade off between IPC organisers introducing new features to PDDL used in the competition, to drive a certain research direction; versus making sure the overall performance of the competitors is assessed in the competition according to more general benchmarks.

I think this raises some interesting discussion points for the workshop, some comments:
1) Getting competition organisers is difficult (especially as they cannot participate themselves) and the opportunity to push research in a certain direction; so restricting their ability to do this, by insisting existing domains have equal (or more) weight, might make recruitment of organisers even more challenging.
2) The number of existing benchmarks is large, so would always easily domainte the new ones, perhaps a selected (hopefully diverse) subset could be taken, rather than all existing domains.
3) I think the idea of categorising and labelling domains to allow better selection of them is definitely a good one, although would need some people willing to dedicate time to it.

---

### Official Review · AnonReviewer3 · 2019-04-26
**Good discussion for the workshop**

**Rating:** 7
**Confidence:** 4

**Review:**

This abstract discusses the impact that IPC's have in the evaluation of research. This is
a relevant topic for the workshop, so it should be accepted to foster discussion around
this topic.

I agree in some of the points raised, specially in that the IPC helps to set standards for
evaluation but it should not be taken as a hard constraint for all planning research. I
also agree that we need to re-design the planning benchmarks. Some comments:

  - "The results of IPCs might sometimes be misinterpreted and misleading in general." It
    is not clear what is claimed here since there is no example. However, I think the
    point to make here is that one cannot draw strong conclusions about what is (and
    specially about what is not!) state of the art solely on the results of an IPC.

  - I disagree about including past domains in the competition. This would allow designing
    "domain-dependent" planners in multiple ways, for example, by overfitting the
    parameters to obtain the best performance on known domains. I do like the fact that
    planner authors do not know anything about the domains their planners will be run on,
    because it is what makes the competition domain independent. Having an additional
    evaluation wouldn't hurt, but it would increase the overhead of running the
    competition.